# Non-Isocyanate Synthesis of Aliphatic Polyurethane by BiCl_3_-Catalyzed Transurethanization Polycondensation

**DOI:** 10.3390/polym16081136

**Published:** 2024-04-18

**Authors:** Bungo Ochiai, Yuriko Kobayashi

**Affiliations:** Graduate School of Science and Engineering, Yamagata University, Yamagata 990-8510, Japan

**Keywords:** polyurethane, bismuth, polycondensation, non-isocyanate

## Abstract

Non-isocyanate polyurethane synthesis by non-Sn catalysis is an essential challenge toward green polyurethane synthesis. Bismuth compounds are attractive candidates due to their low cost, low toxicity, and availability to urethane chemistry. This work applied various Bi catalysts to the self-polycondensation of a bishydroxyurethane monomer and found BiCl_3_ to be an excellent catalyst through optimization. The catalytic activity and price of BiCl_3_ are comparable to those of Bu_2_SnO, while its toxicity is significantly low. BiCl_3_ is, therefore, a promising alternative to Sn-based catalysts in non-isocyanate polyurethane synthesis.

## 1. Introduction

Polyurethanes are applied to a wide range of products because of their widely controllable properties and excellent characteristics, such as foaming ability, biocompatibility, mechanical strength, and abrasion resistance. The industrial fabrication method for polyurethane is the polyaddition of multifunctional isocyanates and diols, developed by Bayer et al. [1,2,3], and is still used nowadays. Because isocyanates are highly reactive, the reaction proceeds quickly and quantitatively yields polyurethanes. However, isocyanates are not only unstable and highly toxic, but their raw materials, phosgene derivatives, are even more poisonous and corrosive. In addition, the reaction of amines with phosgenes generates corrosive hydrogen chloride as a byproduct, which requires the plant facilities to be heavily corrosion-resistant. As a substantial solution to these problems, various green polyurethane syntheses have been developed to prevent the use of phosgenes and/or isocyanates [4,5,6,7,8,9,10,11,12,13,14,15,16,17,18,19,20,21].

Polyaddition of bifunctional five-membered cyclic carbonates obtained by the reaction of diepoxides with carbon dioxide [14,15,22,23,24,25] and diamines is a well explored alternative method [4,5,6,7,8,18,19,20,21]. The resulting polyurethane derivatives bearing hydroxy side chains are called polyhydroxyurethanes, and many studies have been conducted since the patent by Whelan Jr. et al. [4]. However, this method cannot synthesize polyurethanes with the same structure as industrial polyurethanes. Alternatively, polycondensation of dialkylurethanes and bishydroxyurethanes can give such polyurethanes from five-membered cyclic carbonates or linear carbonates and amines without using phosgenes and isocyanates.

Polycondensation of dialkyl urethanes with diols reported by Deepa et al. gives polyurethanes at good yields of 80–90% under solvent-free melting conditions in the presence of titanium-based catalysts, such as Ti(OBu)_4_ [9]. Higher molecular weights are achieved by continuous removal of low-boiling alcohols such as methanol from the polymerization medium under nitrogen purge and subsequent high vacuum. The molecular weight of the resulting polyurethanes ranges from 3000 to 20,000.

CO_2_-based polyurethanes can be obtained by self-polycondensation or transurethane polycondensation with diols of bishydroxyurethanes obtained by the addition reaction of ethylene carbonate (EC) and diamines as monomers, as first reported by Rokicki et al. in 2002 [10]. However, toxic Sn-based catalysts are mainly used because of their high catalytic activities. In addition, efficient removal of ethylene glycol produced as a byproduct requires long reaction times with reduced pressure to synthesize high-molecular-weight polymers [10,11,12,13]. A safer system has been established for similar polymerization under reduced pressure using less toxic BaO [17], titanium alkoxides [13,18], and zinc acetate [13] as catalysts. Various systems have been explored for these non-isocyanate polyurethane syntheses, and polyurethane with higher molecular weight has been developed [15].

For an exploration of new catalysts, we focused on compounds of bismuth, which is an inexpensive and safe heavy element [26,27,28,29]. Many organic and inorganic bismuth compounds have very low toxicity, and some bismuth salts are applied as stomach medicines. Bismuth(III) tricarboxylates, such as bismuth 2-ethylhexanoate and bismuth neo-decanoate, have been developed to replace Sn catalysts for isocyanate-based polyurethane synthesis [30]. The application of BiPh_3_ as a co-catalyst for a Sn catalyst has also been reported [31]. As related applications, bismuth catalysis has been applied for the depolymerization of polyurethanes catalyzed by bismuth neodecanoate [32], transurethanization with alcohols catalyzed by bismuth triflate [33] and reprocessing of cross-linked polyurethane through transurethanization catalyzed by bismuth neodecanoate [34]. Based on these bismuth-catalyzed urethane chemistries, bismuth catalysts were investigated as new catalysts for trans-urethane polymerization. This paper demonstrates that safe BiCl_3_ effectively catalyzes the self-polycondensation of a bishydroxyurethane with an activity comparable to Bu_2_SnO and higher selectivity.

## 2. Materials and Methods

### 2.1. Materials

All the reagents were used as received. EC (>98%), dimethyl sulfoxide (DMSO) super dehydrated (>99.0%, water content < 0.02%), bismuth subsalicylate (Bi composition = 55.6–59.2%), bismuth oxide (>98%), and Bu_2_SnO (>85%) were purchased from Wako Pure Chemical (Osaka, Japan). Dehydrated xylene (>80%, water content < 0.003%), *N*,*N*-dimethylformamide (DMF) super dehydrated (>99.5%, water content < 0.001%), *N*-methyl-2-pyrolidinone (NMP) (>99.0%), and bismuth hydroxide nitrate (residue on ignition as Bi_2_O_3_ = 79.0–82.0%) were purchased from Kanto Chemical (Tokyo, Japan). BiCl_3_ (>97.0%), BiPh_3_ (>98.0%), 1,6-hexanediamine (>99.0%) were purchased from Tokyo Chemical Industry (Tokyo, Japan). BiF_3_ (>99.99%), BiBr_3_ (>98%), Bi(OCOCH_3_)_3_ (>99%), and Bi_2_(SO_4_)_3_ (>90.0%) were purchased from Sigma Aldrich (St. Louis, MO, USA). Bi(OH)_3_ (>90.0%) was purchased from Nakarai Tesque (Kyoto, Japan). The bishydroxyurethane monomer, bis(2-hydroxyethyl)hexane-1,6-diyldicarbamate (BHU6), was prepared as reported from EC and 1,6-hexanediamine (Appendix A) [10,11].

### 2.2. Measurements

^1^H NMR spectra were measured on a JEOL (Tokyo, Japan) ECX-400 (400 MHz) spectrometer using tetramethylsilane as an internal standard. The ^13^C NMR spectrum was measured on a JEOL (Tokyo, Japan) ECX-500 (125 MHz) spectrometer using *d*_6_-DMSO as an internal standard. *d*_6_-DMSO (Kanto Chemical) was used as the solvent. Fourier-transfer infrared (IR) spectra were measured on a Shimadzu (Kyoto, Japan) IRSpirit spectrometer equipped with a Shimadzu QATR-S attenuated total reflection accessory with a diamond disk with a step of 1 cm^–1^. Differential scanning calorimetry (DSC) measurements were performed on a Seiko (Tokyo, Japan) DSC-220 instrument under a nitrogen atmosphere (10 °C min^−1^, N_2_, second heating scan).

### 2.3. Polycondensation of BHU6 (Typical Procedure)

BHU6 (292 mg, 1.00 mmol) and BiCl_3_ (34 mg, 0.11 mmol, 11 mol%) were added to a round-bottomed flask equipped with a side-arm distilling adapter, a Liebig condenser, and a receiver. Then, the apparatus was degassed and purged with nitrogen. Dehydrated xylene (5.0 mL) was added. The mixture was stirred at 150 °C for 5 h under a nitrogen atmosphere. The solid produced in the flask was dissolved in NMP at 100 °C, and the insoluble substance (9.2 mg after drying) was removed by centrifugation at 4100–4200 rpm. The supernatant was poured into a mixed solution of acetone (100 mL) and 1 M HCl aq. (0.1 mL). The resulting [6,2]-polyurethane was collected by filtration and drying under reduced pressure at 60 °C (219 g, yield = 95.1%, *M*_nNMR_ = 5400).

## 3. Results

### 3.1. Effect of Catalyst

Bismuth catalysts (10 mol% to BHU6) were screened for the self-polycondensation of a bishydroxyurethane obtained from the reaction of 1,6-hexanediamine with EC (BHU6) using xylene as a solvent at 145 °C for 3 h under a nitrogen atmosphere (Figure 1, Table 1). The results were compared with a previously reported reaction using Bu_2_SnO as a catalyst [10,11]. The concentration of catalysts was determined according to previous studies that used 5 or 10 mol% of Sn catalysts [10,11]. The molecular weights of the resulting [6,2]-polyurethane were evaluated by ^1^H NMR spectroscopic analysis since [6,2]-polyurethane is insoluble in typical solvents for size exclusion chromatography (SEC) at ambient temperatures. The molecular weights were estimated from the integral ratios of the peaks of the terminal oxyethylene unit and methylene protons adjacent to the nitrogen atom in the repeating unit. Some examples are indicated in the Appendix A. However, this calculation may overestimate the molecular weights in the later stages, where cyclic polymers/oligomers with different structures are produced, as described later.

BiCl_3_ gave polyurethane with lower yield and molecular weight than Bu_2_SnO, while resulting in the highest urethane selectivity among the examined catalysts (Entry 1). BiBr_3_ gave polyurethane with the highest yield and molecular weight among the examined bismuth catalysts (Entry 2). The urethane selectivity was also high. Bismuth subsalicylate was comparable to BiCl_3_ and BiBr_3_ in yield and molecular weight but had a poorer selectivity (Entry 3). Other bismuth compounds resulted in significantly lower yields and selectivity (Entry 4–10). This screening suggests that BiBr_3_ and BiCl_3_ are the superior catalysts among the bismuth catalysts investigated. BiBr_3_ gave polyurethane with the highest molecular weight in the highest yield. BiCl_3_ resulted in the highest selectivity, and its price is approximately 1/5 lower than BiBr_3_. Further optimization studies used BiCl_3_ because of its better performance under the optimized conditions, as described later. Another advantage of BiCl_3_ is its low toxicity. The oral 50% lethal dose (L_D50_) of BiCl_3_ for rats is 3334 mg/kg [35], which is lower than those of Bu_2_SnO (500 mg/kg) [36] BaO (418 mg/kg) [37], zinc acetate (664 mg/kg) [38], and even NaCl (3000 mg/kg) [39], Ti(OBu)_4_ (3122 mg/kg) [40] and Ti(OPr*^i^*)_4_ (7236 mg/kg) [41]. In addition, BiCl_3_ does not exhibit cytotoxicity in human thyroid cancer cells and mitigates the cytotoxicity of gallic acid, probably by complexation [28]. To the best of our knowledge, the toxicity of BiBr_3_ has not been reported.

### 3.2. Mechanism of Catalysis

The catalytic mechanism of BiCl_3_ was investigated by IR spectroscopy (Figure 1 and Appendix A). The IR spectrum of the 2:1 mixture of BHU6 and BiCl_3_ showed *ν*_N-H_ and *ν*_C=O_ shifts. The *ν*_C=O_ peak of BHU6 was observed at 1683 cm^−1^ with a minor shoulder peak at 1651 cm^−1^. The major and shoulder peaks are assigned to hydrogen-bonded ordered and disordered carbonyl groups, respectively [42]. Absorption assignable to free carbonyl groups above 1700 cm^–1^ was unobservable. The even number chain between the urethane groups and symmetrical structure is suitable for intermolecular hydrogen bonding. In the spectrum of the mixture of BHU6 and BiCl_3_, the *ν*_C=O_ absorption clearly shifted to a lower wavenumber with a peak top at 1644 cm^−1^. This shift indicates the strong coordination of the carbonyl group to BiCl_3_, which is more Lewis-acidic than active protons in BHU6. Furthermore, the *ν*_N-H_ peak of BHU6 was observed as a sharp absorption around 3324 cm^−1^, while that of the mixture of BHU6 and BiCl_3_ overlapped with a broad *ν*_O-H_ peak at a higher wavenumber (3376 cm^−1^). This fact also indicates that the N-H proton was released from hydrogen bonding between BHU6 by the interaction between BHU6 and the Lewis acidic BiCl_3_. The strong activation of the carbonyl group by BiCl_3_ is a probable driving force of the nucleophilic substitution. 

Various Bi(III) compounds have been examined for the development of non-toxic and inexpensive catalysts [43,44,45,46,47]. For activation of carbonyl groups by BiCl_3_, the activation of the aldehyde group for an aldol-type addition of a silyl enol ether [43] and Friedel–Crafts reaction via activation of acid halides and anhydrides [44,47] were reported. The activation of the carbonyl group in benzoic anhydride was confirmed by ^13^C NMR spectroscopic analysis [47].

The IR spectrum of a 2:1 mixture of BHU6 and Bu_2_SnO (Figure 1) differs from that of BHU6 and BiCl_3_, implying different mechanisms. In contrast to the BiCl_3_ catalysis, the wavenumbers of the absorption of the *ν*_N-H_ (3326 cm^−1^) and *ν*_C=O(ordered H-bonded)_ (1684 cm^−1^) peaks in the spectrum of the mixture were almost identical to those in the spectrum of BHU6 (3324 and 1683 cm^−1^, respectively). In addition, the absorption of the *ν*_C=O(disordered H-bonded)_ remained observable at 1651 cm^−1^. Thus, Bu_2_SnO negligibly interacted with BHU6. Although the catalytic mechanism for Bu_2_SnO is unclear [48], the transurethanization mechanisms of the Bi- and Sn- catalysis probably differ despite the similar reactivity.

### 3.3. Optimization of Conditions

#### 3.3.1. Effect of Time and Temperature

The polymerization conditions were optimized to attain higher yields and molecular weights. To optimize the polymerization temperature and time, we performed BiCl_3_-catalyzed self-polycondensation of BHU6 at different polymerization temperatures (140, 145, 150, 155, and 160 °C) and polymerization times (1, 3, and 5 h). Before discussing the results of the polycondensation, the aspect of the reaction mixture during the polymerization is described. In the initial stage, the mixture was in a liquid–liquid separation state due to the insolubility of molten BHU6 in xylene at a polymerization temperature higher than the melting point of BHU6. As the polycondensation progressed, solid polyurethane precipitated. In the late stages of the reaction, most parts of polyurethane precipitated, but fewer parts were dissolved in the supernatant.

The polymerization temperature and time were optimized based on yield, conversion, average molecular weight, and the selectivity of the formation of urethane to urea formed by a side reaction. The conversion is the rate of conversion of the hydroxyethyl group to the repeating unit. The time course of conversion was calculated from the integral ratio of the terminal hydroxyethyl group to the repeating unit in the ^1^H NMR spectra of the precipitates before purification, which are the main components of this system. The average molecular weight and the ratio of urethane to urea groups were calculated from the ^1^H NMR spectra of the products obtained by reprecipitation of solutions of both the supernatant and the precipitate in NMP into acetone. The formation of urea groups has been previously shown to occur by the nucleophilic substitution of urethane moieties by amines, which is produced by the tail- and backbiting reaction of the terminal hydroxy group. Figure 2 shows a typical ^1^H NMR spectrum of [6,2]-polyurethane obtained by polymerization at 150 °C for 5 h, and the assignments agreed well with the previous report [11]. The ^13^C NMR spectrum (Appendix A) and the thermal behavior (*T*_g_ = 26 °C and *T*_m_ = 164 °C, Appendix A) are also consistent with [6,2]-polyurethane synthesized by Sn-catalyzed polycondensation and isocyanate-based polyaddition [10,11], supporting the formation of the identical polymer with sufficient purity.

The conversion increased with increasing temperature and time up to 150 °C (Figure 3). On the other hand, at temperatures higher than 150 °C, the conversion rate at 1 h was almost the same as that at 150 °C, but the conversion rate did not increase as time was extended, and the conversion rate at 5 h was lower than the rate at 150 °C.

As with the conversion, the yield increased with increasing temperature and time up to 150 °C. Above 155 °C, the yield at 1 h was comparable to that at 150 °C but did not increase with time, resulting in the highest yield of polyurethane at 150 °C for 5 h (Figure 4).

The number-average molecular weights of the polymers obtained at 1 h tended to increase with temperature (Figure 5). On the other hand, at 3 or 5 h, the molecular weight became highest at 150 °C, and the molecular weights significantly lowered above 155 °C. As a result, the highest molecular weight polyurethane was obtained at 150 °C and 5 h. Yields and molecular weights were correlated with conversions, since the conversion of the terminal groups grew polyurethane with the increase in molecular weight.

The ratios of the urea group were suppressed to within 3% at all temperatures at 1 h. Below 150 °C, the urea ratios slightly increased with time but were within 5% (Figure 6). On the other hand, above 155 °C, the urea ratios increased significantly with increasing time in a similar manner to the previously reported Sn-catalyzed polymerizations [11,49]. As a result of these optimization experiments, polycondensation at 150 °C for 5 h gave the polyurethane with the highest molecular weight and urethane selectivity in the highest yield.

The molecular weights and urea ratios were correlated. In the polymerizations for 1 h, the molecular weight did not decrease even at 160 °C, and the urea ratio stayed lower than 3%. On the other hand, in the polymerizations for 3 and 5 h, molecular weights lowered above 150 °C, and the urea ratios increased significantly from 3–5% at lower temperatures to 7–18%. This relationship suggests that the formation of urea may inhibit the increase in molecular weight.

#### 3.3.2. Discussion on the Mechanism

The results described above are discussed based on the mechanism of this polycondensation (Figure 7). The primary reaction in this system is transurethanization [45], in which the terminal hydroxy group of a monomer or polymer nucleophilically attacks the carbonyl carbon of another monomer or a polymer chain end, and this transurethanization reaction extends the polymer chain (reaction A). On the other hand, ureation occurs as a side reaction in later stages and at higher temperatures [10,11,13,18]. This ureation is initiated by a tail-biting reaction, in which the terminal hydroxy group intramolecularly attacks the urethane carbonyl group, forming a polymer with an amine terminus that is more reactive than the hydroxy terminus and EC (reaction B). A nucleophilic attack of the amine terminus on a urethane group in another polymer chain results in chain extension (reaction C) or disproportionation (reaction D), accompanied by the formation of polymers with urea groups. However, since the average molecular weight remains almost identical after disproportionation, the intermolecular ureation of the amine and urethane hardly interferes with the increase in molecular weight. On the other hand, if the amine terminus generated by backbiting of the hydroxy terminus intramolecularly attacks a urethane carbonyl group and further backbites, this reaction splits the chain into a cyclic oligomer with urethane or urea groups and a linear oligomer with terminal amine or terminal alcohol groups (reaction E). As a result, the polymer chain shortens. Therefore, intramolecular backbiting accompanied by ureation can be assumed to cause the inhibition of molecular weight increase.

Next, the reason why ureation occurred in the later stages of polymerization at high temperatures is discussed. As mentioned above, the solvent and monomer are in a liquid–liquid separation state in the initial stage of polymerization. As the polycondensation proceeds to form polymers with shorter chains, their polarity becomes lower than that of the monomer, and their solubility improves. However, as the polymerization proceeds to form longer polymers, the polymers become insoluble and precipitate out due to entropic disadvantages and increased crystallinity. Since the melting point of the polymer is 150–160 °C, the polymerization temperature below 150 °C, which is lower than the melting point, results in precipitation of the crystallized polymers. Both transurethanization extending polymer chain and ureation shortening polymer chain are suppressed for the precipitated polymers. This crystallization will be an essential factor in inhibiting ureation at low temperatures. In contrast, when the polymerization temperature is above 155 °C, the precipitated polymer is in a molten state and, as a result, the extension of the molecular chain continues. However, in the late stage, the probability of reaction between mostly consumed hydroxy termini becomes significantly lower, and the hydroxy termini backbite to produce amine termini, which causes the scission of polymer chains via ureation. In other words, polycondensation at low temperatures stops the change in molecular weights by precipitation of polymers with sufficient length for crystallinity. In contrast, polycondensation at high temperatures causes a decrease in molecular weight due to continuous reactions of molten terminal groups, including ureation-induced chain scission, in a similar manner to reported polycondensations conducted at temperatures higher than the melting points of polymers [10,11]. 

The formation of urea is a possible factor in the deactivation at higher temperatures. Lewis acid catalysts are not active in a polyurea synthesis [50] via ureation of diurethanes, probably due to the strong coordination of the urea group on the Lewis acids. 

#### 3.3.3. Effect of Solvent

The effect of solvents on the self-polycondensation of BHU6 was investigated using DMF and DMSO, solvents used in isocyanate-based polyurethane synthesis, and anisole, an aromatic ether with a similar boiling point to xylene (Table 2). Polymerization did not proceed in DMF or DMSO (Entry 1 and 2) despite the good solubilities of BHU6 and [6,2]-polyurethane in these polar solvents. A probable reason is the loss of catalytic activity of these highly polar solvents tightly coordinating with Lewis-acidic BiCl_3_. Another possible factor is the high reaction temperature degrading DMF and DMSO to give dimethylamine and disproportionation products of DMSO, which also coordinate with BiCl_3_ to deteriorate its catalytic activity. 

Polycondensation in anisole proceeded through a liquid–liquid separation state in the early stages and produced [6,2]-polyurethane as a precipitate in the late stages (Run 3), as did the polycondensation in xylene. However, the yield, molecular weight, and selectivity of the resulting polyurethane were lower than those of xylene. The polarity of anisole, higher than xylene, while not as high as those of DMF and DMSO, probably resulted in coordination with BiCl_3_, which competes with the activation of BHU6 with BiCl_3_. Therefore, polar solvents are unsuitable for this polycondensation due to the deactivation of BiCl_3_ by coordination, and xylene was the best azeotropic solvent among the solvents examined.

The amount of xylene, which proved to be the best azeotropic solvent, was optimized (Table 3). The appearance of polyurethane obtained at the end of polymerization differed with the solvent amount. Polyurethane obtained using 3 L/mol of xylene was molten (Entry 5), but those obtained using higher amounts of xylene were solid (Entry 1–4). This difference was attributed to the lower molecular weight and urethane selectivity of the polyurethane obtained in Entry 5, which was less crystallizable than the others. This unsuccessful result may be attributed to the insufficient azeotropic removal of ethylene glycol due to the insufficient amount of xylene, which inhibited polycondensation despite the advantages in reaction progression in the molten state. This cessation of transurethanization led to the ureation, and the highest urea ratio in the polyurethane obtained using 3 L/mol of xylene was probably responsible for the synergistic reduction of the melting point.

The increase in solvent volume increased yield, molecular weight, and reaction selectivity. This improvement is attributable to the accelerated azeotropic removal of ethylene glycol, which enhanced the polycondensation. However, further improvement was not achieved by increasing the amount above 5 L/mol. This limitation may originate from the precipitation of the product after a certain degree of growth, which inactivate both the polycondensation and side reactions. This investigation revealed the optimal solvent volume to be 5 L/mol.

#### 3.3.4. Effect of Catalyst Amount

Finally, the amount of catalyst was optimized (Table 4 and Appendix A). Increasing the catalyst amount from 5 mol% to 10 mol% improved yield and molecular weight without deteriorating the selectivity. However, a further increase in catalyst amount to 15 mol% resulted in a slight decrease in yield and molecular weight. This lowered molecular weight is consistent with the higher composition of the urea group, correlating with the scission of the polymer chains.

#### 3.3.5. Polycondensation under Optimized Conditions Using BiCl_3_, BiBr_3_, and Bu_2_SnO

We performed polycondensation of BHU6 under conditions optimized for the BiCl_3_ catalyst system using BiBr_3_ and Bu_2_SnO (Table 5), which gave good results in the catalyst screening described above (Table 1). BiBr_3_ resulted in a lower urethane selectivity than BiCl_3_, as per the conditions mentioned above (Entry 1). The formation of the urea group probably reduced the yield and molecular weight. Bu_2_SnO also had a lower urethane selectivity (Entry 2), although the molecular weight and yield of the polymer were almost comparable to those of BiCl_3_. These number-average molecular weights are higher than those of [6,2]-polyurethane obtained by using zinc acetate (*M*_n_ = 2591) and titanium tetraisopropoxide (*M*_n_ = 863) measured by SEC eluted with hexafluoro-isopropanol [13]. A plausible factor is the higher selectivity of the BiCl_3_ catalysis than the Zn- and Ti-catalysis, which results in higher urea contents (11%) [13]. Since the upper limit of molecular weight attainable in this polycondensation is the molecular weight at which sufficient crystallization occurs, catalysts with adequate activity and selectivity will give comparable results. Thus, BiCl_3_, a Bi catalyst with high selectivity and activity, showed comparable ability to the previously reported Bu_2_SnO.

## 4. Conclusions

This work demonstrated the excellent catalytic activity of BiCl_3_ for the self-polycondensation of BHU6 through the screening of various Bi catalysts and optimization of the conditions. The catalytic activity and price of BiCl_3_ are comparable to those of Bu_2_SnO with high activity among authentic catalysts. Its toxicity is significantly lower than Sn, Zn, and Ba catalysts. The high selectivity and low cost are also advantages. BiCl_3_ is, therefore, a promising green alternative to Sn-based catalysts. The effectiveness of the BiCl_3_ catalyst will contribute to the development of green non-isocyanate polyurethane synthesis by expansion to polycondensation with other hydroxyurethanes and diols.

## Data Availability

All the data substantiating the conclusion of this study are included. Other primary data are available upon reasonable request from the corresponding author.

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
