# Peer review of "Non-Isocyanate Synthesis of Aliphatic Polyurethane by BiCl3-Catalyzed Transurethanization Polycondensation"

_polymers, 2024, doi:10.3390/polym16081136_

Round 1

Reviewer 1 Report

Comments and Suggestions for Authors

-

Dear editor

The manuscript can be improved taking into account the following comments and notes:

1.      The raw materials have not been accurately determined and it is necessary to review the purity and determine the details of the used materials.

2.      In the Materials section, the author reported the used of 1,6-hexanediol. Why?

3.      The author reported the preparation of BHU6 by reaction of EC as a diol by 1,6- Hexanediol as another diol.  "The bishydroxyurethane monomer, bis(2-hydroxy-ethyl)hexane-1,6-diyldicarbamate (BHU6), was prepared as reported from EC and 1,6- Hexanediol [10,11]. " It is not possible. Please edit.

4.      Please describe in the Materials section the details for drying the used Xylene.

5.      The author must exactly identified the concentration of used solutions.

6.      The measurement section have not been accurately described and it is necessary to mentioned the H-NMR solvent and the FT-IR details.

7.      Scheme 1 is not complete. Please provide the total reactions from the beginning of the syntheses.

8.      The assigning of the 1H-NMR of the Fig. 2 needs to be revised. The urethane N-H in this spectrum appears around 4.5-5.0 ppm. Please edit the assignments.(sample ref.: J. APPL. POLYM. SCI. 2017, DOI: 10.1002/APP.44991)

9.      Please provide the 13C-NMR of the polymer.

10.  Please provide the DSC and TGA data for prepared polymer by BiCl3 and compare results with other researcher results.

11.  Please describe why the author select the 5 and 10mol% for experiments.

12.  Table 4. Please provide the H-NMR of the three samples (basic data of the Table 4) and also showed the calculations of the Mn by these spectra.

13.  Also at the end, please describe the advantages of the BiCl3 over the other catalysts apart from the price.

Comments on the Quality of English Language

-

Author Response

Thank you for your constructive comments. We revised our manuscript according to the reviewers' comments. Changes we have made are indicated red in the revised manuscript. The point-by-point responses are indicated below.

  1. The raw materials have not been accurately determined and it is necessary to review the purity and determine the details of the used materials.

The purity of all the reagents were added in the revised manuscript.

  1. In the Materials section, the author reported the used of 1,6-hexanediol. Why?
  2. The author reported the preparation of BHU6 by reaction of EC as a diol by 1,6- Hexanediol as another diol.  "The bishydroxyurethane monomer, bis(2-hydroxy-ethyl)hexane-1,6-diyldicarbamate (BHU6), was prepared as reported from EC and 1,6- Hexanediol [10,11]. " It is not possible. Please edit. 

We are sorry for our careless mistake. We corrected to 1,6-hexanediamine.

  1. Please describe in the Materials section the details for drying the used Xylene. 

Dehydrated xylene was used as purchased. We did not conduct further drying. Water contents of dehydrated reagents were added in the revised manuscript.

  1. The author must exactly identified the concentration of used solutions.

We indicated the amounts of BHU6, catalysts, and solvents for all the Table. The initial solution was heterogeneous, and the exact concentration cannot be calculated. We accordingly could not add the concentrations.

  1. The measurement section have not been accurately described and it is necessary to mentioned the H-NMR solvent and the FT-IR details. 

We added the solvent and the detail of FTIR spectroscopic analysis in the Measurement section. We are sorry for a mistake in the previous version indicating a different spectrometer.

  1. Scheme 1 is not complete. Please provide the total reactions from the beginning of the syntheses. 

The polymerization was conducted for the purified authentic BHU6. I think that typical schemes for polymerizations using known monomers synthesized does not contain their monomer syntheses. We accordingly did not add the monomer synthesis in Scheme 1 to avoid confusion that this polymerization was conducted in one-pot or one-step and that the monomer is novel.

  1. The assigning of the 1H-NMR of the Fig. 2 needs to be revised. The urethane N-H in this spectrum appears around 4.5-5.0 ppm. Please edit the assignments.(sample ref.: J. APPL. POLYM. SCI. 2017, DOI: 10.1002/APP.44991)

The solvent for the measurement is different, but I made a mistake that the terminal urethane and urea protons were indicated reversely. The corrected assignment agrees with the previous report [10 and 11]. The calculation was conducted by the correct assignment and the results are identical. For a reference, we added the spectrum of the monomer in Figure S1, which is also consistent with the previous reports. The urethane proton more acidic (namely in the lower magnetic field region) than the OH proton was observed at 6.8 and 7.2 ppm due to the presence of s-cis and s-trans tautomers in DMSO. Our polymer is insoluble in CDCl3 and the comparison with the indicated reference was impossible.

  1. Please provide the 13C-NMR of the polymer. 

We added the 13C NMR spectrum in Figure S2.

  1. Please provide the DSC and TGA data for prepared polymer by BiCl3 and compare results with other researcher results. 

The DSC curve was added as Figure S3. The Tg and Tm are consisted with the previously reported [6,2]-polyurethane prepared by Sn-catalyzed polycondensation and isocyanate-based polyaddition. This explanation was added at the last of section 3.3.1.

  1. Please describe why the author select the 5 and 10mol% for experiments. 

The first report of the similar polycondensation by Rokicki [10] used 5 mol% of Sn catalysts, and our previous work used 5 and 10 mol% of catalysts. We accordingly started with the similar amounts. The amounts in the previous studies were added in the section 3.1

  1. Table 4. Please provide the H-NMR of the three samples (basic data of the Table 4) and also showed the calculations of the Mn by these spectra. 

The calculation method was added in Section 3.1 with the possibility of the overestimation in the latter stage. The original data for Table 4 was added as Figure S5.

  1. Also at the end, please describe the advantages of the BiCl3 over the other catalysts apart from the price.

The safety and selectivity are also important. We revised the section 3.1 and conclusion to emphasize the safety of BiCl3.

Reviewer 2 Report

Comments and Suggestions for Authors

The manuscript was good enough in terms of presentation and methodology. Moreover, the paper is subjected to major improvement.

1.     Page 3 line 110-112: ‘We found BiBr3 and BiCl3 to be the superior catalysts among the bismuth catalysts investigated, by the highest yield and molecular weight’ è was not highest.

2.     Page 3 line 110-113: correct the sentence.

3.     For further optimization studies, we used BiCl3 because of its better performance under the optimized conditions, as described later  è according to table 1 BiBr3 showed the better performance (yield = 74%, Mn =2100) compare to BiCl3 (yield=59%, Mn=1600). Need to state clearly the reason to choose BiCl3 instead of BIBr3.

4.     Describe the importance of urethane/Urea ratio. According to table 1, Bu2SnO made highest Mn (2400) and urea 14%  but using BrCl3, Mn was 1600 and urea 3% è how can explain this correlation? Normally higher urea% reduced the Mn.

5.     Try to avoid using first persons “we……”.

6.     What was the reasons for decreasing conversion at temperature above 150-degree C?

7.     Give the proper reference while describe the mechanism.

8.     Reference are missing in all evaluation of causes in discussion.

9.     Compare your results with literature.

10.  In table 2-5, the yield for BiCl3 catalyst and Xylene as solvent was more than 90% and Mn was around 5000 but in table it was 59% and 1600 respectively. Need explanation.  

Comments on the Quality of English Language

Moderate editing of English language required

Author Response

Thank you for your constructive comments. We revised our manuscript according to the reviewers' comments. Changes we have made are indicated red in the revised manuscript. The point-by-point responses are indicated below.

  1. Page 3 line 110-112: ‘We found BiBr3 and BiCl3 to be the superior catalysts among the bismuth catalysts investigated, by the highest yield and molecular weight’ è was not highest. 
  2. Page 3 line 110-113: correct the sentence. 

We clarified the sentence by splitting sentences as follows.

This screening suggests BiBr3 and BiCl3 to be the superior catalysts among the bismuth catalysts investigated. BiBr3 gave polyurethane with the highest molecular weight in the highest yield. BiCl3 resulted in the highest selectivity, and its price is approximately 1/5 lower than BiBr3.

  1. For further optimization studies, we used BiCl3 because of its better performance under the optimized conditions, as described later  è according to table 1 BiBr3 showed the better performance (yield = 74%, Mn =2100) compare to BiCl3 (yield=59%, Mn=1600). Need to state clearly the reason to choose BiCl3 instead of BIBr3. 

      The reason was emphasized. The cost and safety are the advantages of BiCl3. BiCl3 is a very safe compound as added at the last part of Section 3.1. In contrast, the risk of BiBr3 has not been reported to the best of our knowledge. 

  1. Describe the importance of urethane/Urea ratio. According to table 1, Bu2SnO made highest Mn (2400) and urea 14%  but using BrCl3, Mn was 1600 and urea 3%. How can explain this correlation? Normally higher urea% reduced the Mn. 

      Conversion is the major factor for the molecular weight. Formation of urea, which proceeds in the latter stage, decreases the molecular weight, but the effect is weaker than the conversion. As indicated in Figure 3, the conversions in the reaction at 145 ËšC are low.

      We revised the texts to clarify that Section 3.1 is for initial catalyst screening and that the later sections are for optimization.

  1. Try to avoid using first persons “we……”.

      We revised all the sentences.

  1. What was the reasons for decreasing conversion at temperature above 150-degree C?

      Ureas are strong ligands for Lewis acids. Lewis acid catalysts are inactive in ureation of urethanes with amines. Similar deactivation is a plausible reason for the lower conversions at high temperatures. This discussion was added in Section 3.3.2.

  1. Give the proper reference while describe the mechanism.
  2. Reference are missing in all evaluation of causes in discussion. 

      We added discussion with references for catalysts (Section 3.1), polymer structures (Section 3.3.1), mechanism (Section 3.3.2), and the catalytic efficiency (3.3.5) in addition to the mechanism.

  1. Compare your results with literature.

      The thermal behavior consistent with the previous work was added in Section 3.3.1 and Figure S3. The molecular weights attained in this work are higher than a previous work on Ti- and Zn-catalyzed polycondensation using the same monomers probably due to the higher selectivity [added reference 13]. This comparison was added in Section 3.3.5.

  1. In table 2-5, the yield for BiCl3 catalyst and Xylene as solvent was more than 90% and Mn was around 5000 but in table it was 59% and 1600 respectively. Need explanation.  

The data in the first table is the result of the catalyst screening before the optimization studies. We added a sentence " The polymerization conditions were optimized to attain higher yields and molecular weights." at the beginning of Section 3.3.

Reviewer 3 Report

Comments and Suggestions for Authors

Non-isocyanate synthesis of aliphatic polyurethane by BiCl3 catalyzed transurethanization polycondensation

In the manuscript titled “Non-isocyanate synthesis of aliphatic polyurethane by BiCl3 catalyzed transurethanization polycondensation”, the researchers have synthesized Non-isocyanate polyurethane synthesis by non-Sn catalysis method. Considering the advantages of Bismuth, BiCl3 was found to be a promising alternative to Sn-based catalysts in non-isocyanate polyurethane synthesis.

The following points shall be addressed by the authors before considering this paper for publication.

Major issues

1.      In the introduction, the authors have added that However, the four steps of a longer process and the need for reduced pressure are still issues that need to be solved. How far does the authors authenticate that the issues are addressed in this study?.

2.      The authors shall outline why toxic Sn-based catalysts are used in the synthesis of polyurethane? Is there any other alternative? How is safety ensured? What are the potential dangers when using Bismuth?

3.      In Fig 1, (a), the authors shall mark the wave numbers?. Also the authors shall compare them with those in literature.

4.      Results and Discussion part is quite weak and needs to be strengthened. Also no valid discussion is made

5.      What is ureation? What is backbiting reaction? Authors shall shed light on these concepts.

6.      From authors point of view what is the strength of this research and how does this research advance the filed?

7.      Most of the references are quite old. How do the findings of this study expose the research area?

8.      What is the present development in this field?

Comments on the Quality of English Language

The following sentence shall be revised to deliver correct meaning.

1.      The high boiling point of ethylene glycol, produced as a byproduct, makes sufficient removal difficult, and long reaction times with reduced pressure are required to synthesize high-molecular-weight polymers.

2.      The number-average molecular weight tended to increase with temperature at 1 h (Figure 5)..

English polishing can improve the paper.

Author Response

Thank you for your constructive comments. We revised our manuscript according to the reviewers' comments. Changes we have made are indicated red in the revised manuscript. The point-by-point responses are indicated below.

  1. In the introduction, the authors have added that However, the four steps of a longer process and the need for reduced pressure are still issues that need to be solved. How far does the authors authenticate that the issues are addressed in this study?.

We agree that this sentence is inappropriate. We removed this sentence. Instead, we added an explanation that this work using EC as a resource for the monomer is CO2-based. 

  1. The authors shall outline why toxic Sn-based catalysts are used in the synthesis of polyurethane? Is there any other alternative? How is safety ensured? What are the potential dangers when using Bismuth?

 The explanation of toxicity of bismuth was improved (Section 3.1). Although the toxicities of bismuth compounds are not fully revealed, various bismuth compounds including BiCl3 have toxicity lower than NaCl.

  1. In Fig 1, (a), the authors shall mark the wavenumbers?. Also the authors shall compare them with those in literature.

We added Figure S2 indicating full-range FTIR spectra with some marks on the spectra, since the addition of marks on Figure 1 made the visibility lower. We resurveyed research on mechanism of BiCl3-catalysis and found 2 papers added in the revised version [43 and 47]. Ref 43 does not show detail, but ref [47] shows evidence of coordination by NMR. 

  1. Results and Discussion part is quite weak and needs to be strengthened. Also no valid discussion is made

      We added discussion with references for catalysts (Section 3.1), polymer structures (Section 3.3.1), mechanism (Section 3.3.2), and the catalytic efficiency (3.3.5) in addition to the mechanism.

  1. What is ureation? What is backbiting reaction? Authors shall shed light on these concepts.

      Discussion of Section 3.3.2. and Figure 7 were improved.

  1. From authors point of view what is the strength of this research and how does this research advance the filed?

We improved the conclusion based on the improved discussion. BiCl3 is the catalyst meeting activity, low toxicity, low cost, and selectivity.  

  1. Most of the references are quite old. How do the findings of this study expose the research area?

12 literature references were added in the revised manuscript. We retained the old original references since the initial works are also important. 

  1. What is the present development in this field?

As indicated in the reply to Comment 6, the effectiveness of BiCl3 was emphasized in the revised manuscript.

Comments on the Quality of English Language

The following sentence shall be revised to deliver correct meaning.

  1. The high boiling point of ethylene glycol, produced as a byproduct, makes sufficient removal difficult, and long reaction times with reduced pressure are required to synthesize high-molecular-weight polymers.

 We corrected the sentence to

In addition, efficient removal of ethylene glycol produced as a byproduct requires long reaction times with reduced pressure to synthesize high-molecular-weight polymers.

  1. The number-average molecular weight tended to increase with temperature at 1 h (Figure 5).

  We corrected the sentence to

The number-average molecular weights of the polymers obtained at 1 h tended to increase with temperature.

Reviewer 4 Report

Comments and Suggestions for Authors

This manuscript is a research article, reporting  Bi compounds as catalysts to the self-polycondensation of a non-isocyanate polyurethane synthesis. This manuscript is well written and this work is well planned and of highly scientific standard, so it should be accepted for publication in Polymers, after a minor suggestions, as follows:

(1) Throughout the manuscript, it is recommended to use narrative structures, so I suggest to rewrite the sentences that contain "We ..." in lines 10, 66, 98, 110, 113, 121, 154, 160, 213, 234, 256, 278, 294, 304, and 327.

(2) Remove "furthermore" in line 24.

Author Response

Thank you for your comments. We revised our manuscript according to the reviewers' comments. Changes we have made are indicated red in the revised manuscript. The point-by-point responses are indicated below.

(1) Throughout the manuscript, it is recommended to use narrative structures, so I suggest to rewrite the sentences that contain "We ..." in lines 10, 66, 98, 110, 113, 121, 154, 160, 213, 234, 256, 278, 294, 304, and 327.

(2) Remove "furthermore" in line 24.

Thank you for your kind instruction. We revised our manuscript according to your suggestion. Changes we have made are indicated in red in the revised manuscript.

Round 2

Reviewer 1 Report

Comments and Suggestions for Authors

-

Reviewer 2 Report

Comments and Suggestions for Authors

The revised manuscript has improved a lot.

Comments on the Quality of English Language

Minor editing of English language required